# $^{31}$P NMR Investigations on Roundup Degradation by AOP Procedures

**Marcin H. Kudzin [1,*], Renata Żyłła [1] , Zdzisława Mrozińska [1] and Paweł Urbaniak [2]**

[1] Textile Research Institute, Brzezinska 5/15, Lodz 92-103, Poland; zylla@iw.lodz.pl (R.Ż.); zmrozinska@iw.lodz.pl (Z.M.)

[2] Faculty of Chemistry, University of Lodz, Tamka 12, Lodz 90-136, Poland; pawelurb@chemia.uni.lodz.pl

\* Correspondence: kudzin@iw.lodz.pl; Tel.: +48-42-6163-100

**Abstract:** The reactions of (*N*-(PhosphonoMethyl)Glycine) PMG with $H_2O_2$ in homogenous systems were investigated using $^{31}$P NMR (Nuclear Magnetic Resonance). These reactions were carried out in two reaction modes: without UV radiation and under UV radiation. The reactions of PMG with $H_2O_2$ without UV radiation were carried out in two modes: the degradations of PMG (0.1 mmol) by means of 5–10 molar excess of hydrogen dioxide (PMG-$H_2O_2$ = 1:5 and 1:10) and the degradation of PMG (0.1 mmol) in homogenous Fenton reactions (PMG-$H_2O_2$-$Fe^{2+}$ = 1:10:0.05 and 1:10:0.1). All reactions were carried out at ambient temperature, at pH 3.5, for 48 h. The reactions of PMG (in Roundup herbicide composition, 12 mmol) with $H_2O_2$ under UV radiation (254 nm) were carried out using $5 \times$ molar excess of $H_2O_2$ (60 mmol), in the pH range of $2 \leq$ pH $\leq 12$, for 6 h. In this mode of PMG oxidation, the splitting of C-P was observed in the ratios dependent on the applied pH of the reaction mixture.

**Keywords:** glyphosate; oxidation stability; oxidative dephosphonylation; Fenton reaction

## 1. Introduction

Glyphosate (*N*-(PhosphonoMethyl)Glycine) (PMG) is a broad-spectrum systemic herbicide, invented by Franz and brought to market in 1974 by Monsanto under the trade name Roundup [1,2].



**PMG**

Glyphosate, absorbed mainly through foliage, inhibits a plant enzyme—5-enolpyruvylshikimate-3-phosphate (EPSP) synthetase (EC 2.5.1.19)—involved in the bio-synthesis of aromatic amino acids (AAA) substrates in a build-up of plant lignins [3–5].

A significant fraction of glyphosate sprayed on crops returns to the soil, and in spite of strong adsorption to soil solids [6], is able to contaminate surface water (runoff and erosion) [7–9].

Animal and epidemiological studies published in recent decades point to a potential glyphosate toxicity [10–12]. Further, the World Health Organization's International Agency for Research on Cancer concluded that glyphosate is "probably carcinogenic to humans" [13] and Anifandis et al. [14,15] demonstrated that glyphosate/PMG induce DNA fragmentation.

Therefore, various treatment processes have been investigated to reduce the pesticide concentration in water and to minimize the potential health risks associated with exposure to these

chemicals by the consumption of contaminated water [16–26]. Advanced oxidation processes (AOPs) are a key technology to solve pesticide contamination problems during both water and wastewater treatment (Table 1).

**Table 1.** Representative applications of advanced oxidation process (AOP) technology in the chemical degradation of *N*-(PhosphonoMethyl)Glycine (PMG) and aminomethylphosphonic acid ($Gly^P$).

| No | P-Herbicide | AOP Systems | Phosphorus Degradation Products | Analysis | Reference |
|----|-------------|-------------|----------------------------------|----------|-----------|
| 1 | PMG and $Gly^P$ | Birnessite ($Mn^{4+}$ and $Mn^{3+}$) | $H_3PO_4$ | VIS (P-Mo Blue) | [20] |
| 2 | PMG | $Fe^{(III)}(C_2O_4)_n{}^{m-}$/UV (365 nm) | $H_3PO_4$ | VIS (P-Mo Blue) | [21] |
| 3 | PMG | $TiO_2$/UV (312 nm) | $H_3PO_4$ | HPLC-UV | [22] |
| 4 | PMG | $O_2$/$TiO_2$/$SiO_2$/UV (365 nm) | $H_3PO_4$ | TOC, HPLC | [23] |
| 5 | PMG and $Gly^P$ | $O_3$ (pH = 6.5 and 10) Photolysis (292 nm) $TiO_2$/$O_2$ (292 nm) | $H_3PO_4$ $H_3PO_4$ $H_3PO_4$ | TOC, HPLC-FD TOC, HPLC-FD TOC, HPLC-FD | [24] |
| 6 | PMG | $H_2O_2$/UV (254 nm) | $H_3PO_4$ | HPLC-FD | [25] |
| 7 | PMG | Birnessite ($Mn^{4+}$ and $Mn^{3+}$) | $H_3PO_4$ + $Gly^P$ + C-$P^x$ | $^{31}P$ NMR | [26] |

Birnessite $(Na_{0.3}Ca_{0.1}K_{0.1})(Mn^{4+}, Mn^{3+})_2O_4 \times 1.5\ H_2O$. $Fe^{(III)}(C_2O_4)_n{}^{m-} - Fe(C_2O_4)_2{}^-$ and $Fe(C_2O_4)_3{}^{3-}$. $P_i$ colorimetrical determination using the phospho-molybdate blue reaction. TOC—total carbon. C-$P^x$—unidentified compound.

The various mechanisms of PMG degradation using AOP technology, according to the literature, were presented by Manassero [25]. These not always coherent results (Figure 1) led us to investigate these process by using $^{31}P$ NMR monitoring of the PMG degradation processes. This $^{31}P$ NMR technique has been applied for the analysis of PMG metabolites and degradation products in only a few earlier research works [26–33].

**Figure 1.** Degradation products of *N*-PhosphonoMethyl)Glycine (PMG) of representative methods.

## 2. Materials and Methods

### 2.1. Materials

The phosphonic amino acids used in the studies (Table 2) were synthesized as follows: aminomethylphosphonic acid ($Gly^P$) was synthesized according to the Soroka method [34,35]; (*N*-methylamino)methylphosphonic acid (Me-$Gly^P$) was obtained by the hydrophosphonylation of 1,3,5-trimethylhexahydro-1,3,5-triazine by means of diisopropylphosphite according to

Maier [36]; (*N,N*-dimethylamino)-methyl-phosphonic acid (Me$_2$-Gly$^P$) was obtained by the modified Kabachnik-Fields condensation [37,38].

**Table 2.** Names, abbreviations, and structures of aminophosphonic acids discussed in this work [a].

| Structure | | Name | Trivial Name | Abbr. | |
|---|---|---|---|---|---|
| $\begin{array}{c}R^1\\ \phantom{x}\diagdown\phantom{xx}O\\ \phantom{xxx}\parallel\\ N-C-P(OH)_2\\ R^{2}\diagup\phantom{x}H_2\end{array}$ | | 1-aminoalkylphosphonic acid | phosphono amino acid | AA$^P$ | Synth. |
| No | R | R$^1$ | | | |
| 1 | H | H | aminomethylphosphonic acid | phosphono-glycine | Gly$^P$ | [35] |
| | Me | H | (*N*-methylamino)methyl-phosphonic acid | phosphono-(*N*-methyl)-glycine | Me-Gly$^P$ | [37] |
| | Me | Me | (*N,N*-dimethylamino)-methylphosphonic acid | phosphono-(*N,N*-dimethyl)-glycine | Me$_2$-Gly$^P$ | |

[a] Applied names are in accordance with the IUPAC rules, and abbreviations are in agreement with the general rules elaborated by [38,39].

Phosphonomethyl glycine, methylphosphonic acid, 1,3,5-trimethylhexahydro-1,3,5-triazine, catalase and other applied reagents and solvents were purchased from Aldrich. Diisopropylphosphite was purchased from ACROS Organic$^{TM}$.

Herbicide Roundup Ultra 170 SL, containing glyphosate-isopropylammonium salt [227.2] (CAS: 38641-94-0; 170 g/L; 15.67%; 0.75 M), and surfactant (CAS not given; 8%) were purchased from Monsanto Europe S.A (Scotts Poland, Warsaw, Poland).

*2.2. Synthesis of Aminophosphonic Standards*

2.2.1. Synthesis of Aminomethylphosphopnic Acid (Gly$^P$)

Phosphorus chloride(III) (8.75 mL; 0.10 mol) was added dropwise to a well-stirred mixture of *N*-(hydroxymethyl)benzamide (synthesized by the hydroxymethylation of benzamide [34]) (15.1 g; 0.10 mol) and anhydrous acetic acid (20 mL), at ambient temperature. The mixture was then refluxed for 1 h, evaporated (25 °C at 10–20 mm Hg for 15 min, and 75 °C at 0.05 mm Hg) to an oily residue and dissolved in hydrochloric acid solution (5 M aq.; 100 mL). The mixture was heated under reflux for 8 h, cooled to room temperature, and the separated benzoic acid was filtered off. The filtrate was evaporated, the residue was dissolved in water (20 mL), then the solution of crude Gly$^P$ was purified on a Dowex 50 W × 8 ion exchange column using water elution. The obtained Gly$^P$ (8.1 g; 0.072 mol; 72%) was homogeneous at $^{31}$P NMR solutions. Elemental analysis data (determined %/(calculated %)) for CH$_6$NPO$_3$ [111.04] C = 10.70 (10.82); H = 5.47 (5.45); N = 12.50 (12.61).

2.2.2. Synthesis of *N*-Methylaminomethylphosphopnic Acid (Me-Gly$^P$)

To trimethylhexahydro-s-triazine (1.29 g; 0.01 mol diisopropylphosphite (5.0 g; 0.03 mol), was added and the mixture was heated with stirring to 100–110 °C for 4 h. The reaction mixture was evaporated (25 °C at 10–20 mm Hg for 15 min, and 75 °C at 0.05 mm Hg), then diluted with 5 M HCl (100 mL) and refluxed for 8 h. The hydrolyzate was evaporated to an oily residue, which was dissolved in water (20 mL) and extracted with ethyl ether (20 mL). The aqueous layer was passed through a Dowex 50 W × 8 ion exchange column using water elution. Fractions were collected (molybdate test) and evaporated. The crystalline product was washed with ethanol, filtered, and dried to give 2.0 g (53.3%) of Me-Gly$^P$, homogeneous $^{31}$P NMR solutions. Elemental analysis data (determined %/(calculated %)) for C$_2$H$_8$NPO$_3$ [125.06]: C = 19.12 (19.21); H = 6.55 (6.45); N = 11.10 (11.2).

### 2.2.3. Synthesis of *N,N*-Dimethylaminomethylphosphopnic Acid (Me$_2$-Gly$^P$)

The formaldehyde aqueous solution (37%; d = 1.11 g/mL; 2.9 g; 0.05 mol) was gradually added to a stirred mixture of equimolar quantities of dimethylamine (2 M solution of Me$_2$NH in methanol; 25 mL; 0.05 mol) and diethyl phosphite (7.8 g; 0.05 mol), keeping the temperature below 85 °C. The reaction mixture was stirred for 30 min, and evaporated (25 °C at 10–20 mm Hg for 15 min, and 75 °C at 0.05 mm Hg) to an oily residue. The residue was dissolved in 5 M HCl (100 mL) and the solution was refluxed for 8 h. The hydrolyzate was evaporated to dryness (60 °C; 10–20 mm), and the residue was passed through a Dowex 50 W × 8 ion exchange column using water elution. The collected fractions (phosphomolybdate assay) were evaporated to dryness giving white crystals of Me$_2$Gly$^P$ (4.2 g; 60.0%), homogeneous in$^{31}$P NMR solutions. Elemental analysis data (determined %/(calculated %)) for C$_3$H$_{10}$NPO$_3$ [139.09]: C = 25.78 (25.91); H = 7.32 (7.25); N = 9.98 (10.07).

### 2.2.4. Solutions

- Solution of 0.01 M Fe(II): a sample of FeSO$_4$ × 7H$_2$O [278] (28 mg) was dissolved in water (10 mL).
- Solution of 0.02 M Fe(II): a sample of FeSO$_4$ × 7H$_2$O [278] (56 mg) was dissolved in water (10 mL).
- Catalase solution: a sample of 10 mg of catalase was dissolved in 50 mL of distilled or deionized water.
- Solution of 2 M H$_2$SO$_4$ (in 20% D$_2$O): samples of 2.5 M H$_2$SO$_4$ (20 mL) were diluted to 25 ml with D$_2$O (5 mL).

### 2.3. Instruments

$^{31}$P NMR spectra were recorded on a Bruker AC 200 spectrometer operating at 81.01 MHz and on a Bruker Avance III 600 spectrometer operating at 242.9 MHz. $^1$H NMR spectra were recorded on a Bruker Avance III 600 spectrometer operating at 600 Hz. Positive chemical shift values of $^{31}$P were reported for compounds absorbed at lower fields than H$_3$PO$_4$.

The pH measurements were performed using a CX-505 multifunction laboratory meter (Elmetron, Zabrze, Poland) equipped with a combination pH electrode EPP-1 (Elmetron, Zabrze, Poland). The chemical degradation of aqueous solutions of the Roundup herbicide formulation in PMG-H$_2$O$_2$ and PMG-H$_2$O$_2$-UV systems was carried out. Experiments were performed in the reactor shown in Figure 2.

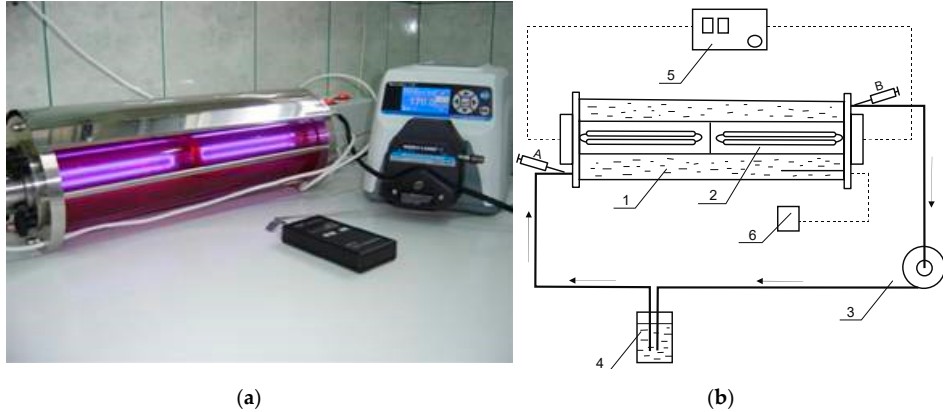

(**a**)　　　　　　　　　　　(**b**)

**Figure 2.** Reactor used for Roundup degradation by means of AOP technology. (**a**) Figure of a photoreactor with UV lamp used for the oxidation of PMG by means of UV/H$_2$O$_2$. (**b**) Schematic diagram: 1—glass reactor; 2—quartz tube with UV lamp (254 nm; 22 W); 3—peristaltic pump; 4—reactor connector; 5—UV power supply; 6—temperature detector; A, B—samples. Applied conditions: PMG (12 mmol), H$_2$O$_2$ (60 mol) in an aq. solution (3.7 L) at different initial pH values applied: 2 ≤ pHs ≤ 12. Irradiation time was up to 360 min. Temperature was 25 °C.

*2.4. Degradation of Glyphosate*

2.4.1. Degradation of Phosphonomethyl Glycine (PMG)

Samples of PMG (0.1 mmol) were dissolved in appropriate volumes of $FeSO_4$ solution (0.5 mL of 0.01 M/0.02 M solution of $FeSO_4$), followed by the addition of dihydrogen dioxide (the exact amounts of the solutions are given in Table 3) and kept at room temperature for a desired reaction time. Then, the reaction mixtures were centrifuged, acidified with 2 M $H_2SO_4$ to pH = 3.5 if necessary, and the volumes of 0.4 mL were taken and mixed with $D_2O$ (0.05 mL) and 0.1 M EDTA (0.05 mL), and analyzed by means of $^{31}P$ NMR.

**Table 3.** Preparation of the reaction mixtures for PMG-$H_2O_2$ and PMG-$H_2O_2$-$Fe^{2+}$ systems.

| Exp. | PMG 0.1 mmol | $H_2O$ | 10 M $H_2O_2$ | | $FeSO_4$ | |
|------|------|------|------|------|------|------|
| | | | 0.5 mmol | 1 mmol | 0.01 M | 0.02 M |
| 1 | 17 mg | 0.55 mL | 0.05 mL | – | – | – |
| 2 | 17 mg | 0.5 mL | – | 0.1 mL | – | – |
| 3 | 17 mg | – | – | 0.1 mL | 0.5 mL (0.005 mmol) | – |
| 4 | 17 mg | – | – | 0.1 mL | – | 0.5 mL (0.01 mmol) |

2.4.2. Degradation of Roundup Herbicide Formulation by Means of AOP Technology

Reaction mixtures were prepared in accordance with Table 4. Thus, samples of PMG contained in Roundup Ultra 170 SL Herbicide solution (0.75 M; 16 mL; 12 mmol of PMG), were diluted in water (3680 mL), dihydrogen dioxide samples (60 mmol) were added, and the reaction mixtures were adjusted to the desired pH value by means of acidification with 2 M $H_2SO_4$ or alkalized by means of 5 M KOH. The oxidative degradations of herbicide were performed for the desired time, during which the reaction progress was monitored by the $^{31}P$ NMR analysis. Thus, the appropriate samples (4 mL) were treated with catalase (0.1 mL), kept for 30 min at room temperature, and evaporated to an oily residue. These were dissolved in 2 M $H_2SO_4$ (in 20% $D_2O$) (0.5 mL) and analyzed using $^{31}P$ NMR.

**Table 4.** Preparation of the reaction mixtures for AOP degradations of PMG in Roundup herbicide.

| No | Roundup (0.75 M) | | $H_2O$ | $H_2O_2$ (30%; 10 M) | | $H_2SO_4$ (2 M) | NaOH (5 M) | pH |
|----|------|------|------|------|------|------|------|------|
| | mL | PMG mmoL | mL | mL | mmoL | mL | mL | |
| 1 | 16.0 | 12.0 | 3 680 | – | – | – | – | 4.85 |
| 2 | 16.0 | 12.0 | 3 680 | 6.0 | 60.0 | 4.6 | – | 2.0 |
| 3 | 16.0 | 12.0 | 3 680 | 6.0 | 60.0 | 0.5 | – | 4.0 |
| 4 | 16.0 | 12.0 | 3 680 | 6.0 | 60.0 | – | 4.0 | 6.0 |
| 5 | 16.0 | 12.0 | 3 680 | 6.0 | 60.0 | – | 4.8 | 8.0 |
| 6 | 16.0 | 12.0 | 3 680 | 6.0 | 60.0 | – | 6.0 | 10.0 |
| 7 | 16.0 | 12.0 | 3 680 | 6.0 | 60.0 | – | 40.0 | 12.0 |

## 3. Results and Discussion

*3.1. Protonation Equilibria of Reagents*

Representative values of p$K_a$ are given in Table 5. On this basis, protonation equilibria of phosphonomethyl glycine (PMG) are presented in Figure 3.

**Table 5.** Representative works on pKa determination of PMG.

| No. | pK | | | | Method | Reference |
| | $pK_1$ | $pK_2$ | $pK_3$ | $pK_4$ | | |
|---|---|---|---|---|---|---|
| 1 | 2.0 | 2.6 | 5.6 | 10.6 | pH metric titration | [40] |
| 2 | | 2.32 | 5.86 | 10.86 | pH metric titration | [41] |
| 3 | <1 | 2.0 | 5.5 | 10.5 | $^1$H and $^{31}$P NMR | [42] |
| 4 | 0.3 | 2.3 | 5.6 | 10.6 | $^1$H and $^{31}$P NMR | [43] |
| 5 | | 2.09 | 5.52 | 10.28 | pH metric titration | [44] |
| 6 | logβ 9.66 (1.58) | logβ 14.86 (5.20) | log β 16.44 (9.66) | | pH metric titration | [45] |

**Figure 3.** Dissociation/protonation equilibria of glyphosate (lower branch (arrows in magenta) considering the reports of Peixoto et al., 2015 [45] and Liu et al., 2016 [46]).

Despite the large amount of research data, there is conflicting information in the literature concerning the first protonation step of glyphosate, namely whether the first dissociable proton in the $HL^{2-}$ formations is attached to the nitrogen atom of the amino group or to the oxide atom of the phosphonate function. As a matter of fact, only two recent reports consider that the first protonation step occurs on the one of the oxygen atoms in the phosphonate group (Figure 4, magenta arrows) [45,46].

On the basis of this analysis of the speciation graph (Figure 4), we assumed that in aqueous solutions PMG exists in the following forms: at pH = 0—mainly as $H_4L^+$ form; at pH = 1.5—mainly as $H_3L$ form; at pH = 3.5–4—mainly as $H_2L^-$; at pH = 8—mainly as $HL^{2-}$; and at pH ≥ 12—mainly as $L^{3-}$ form.

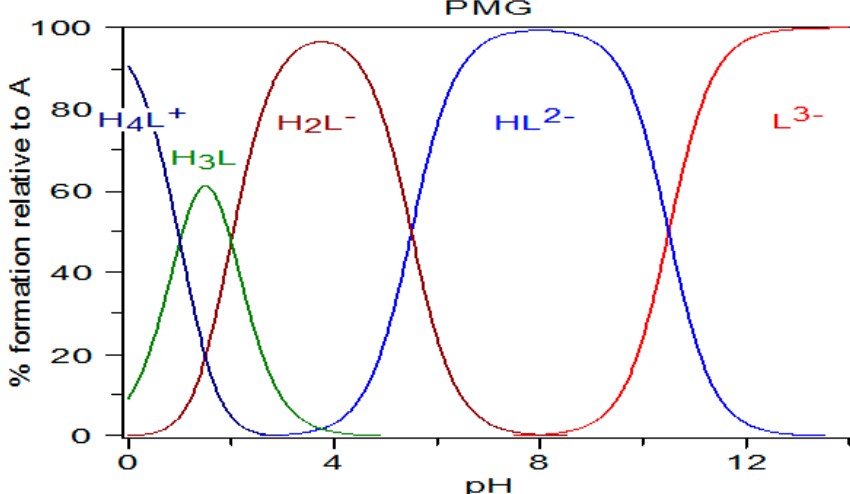

**Figure 4.** Diagram of PMG species distribution calculated using the $pK_a$ values of Appleton et al. [42] ($pK_{a1} < 1$; $pK_{a2} = 2.0$; $pK_{a3} = 5.5$; $pK_{a4} = 10.5$), and the HySS program (Alderighi et al., 1999) [47].

The protonation equilibrium of dihydrogen dioxide is shown in Figure 5; dihydrogen dioxide speciation is also presented in Figure 6. In the literature, the pKa values for $H_2O_2$ are as follows: $pK_{a1} = -3.1$ [48] and $pK_{a2} = 11.6$ [49]. This means that in concentrated $H_2SO_4$ (e.g., 2 M), dihydrogen dioxide can exist in the $H_3L^+$ form, at pH = 0–8 it exists in the molecular form $H_2L$, at pH = 14 it is dissociated in ca. 50% to $HL^-$, and for 2 M KOH (pH > 14) it is almost fully ionized.

**Figure 5.** Dissociation/protonation equilibria of dihydrogen dioxide.

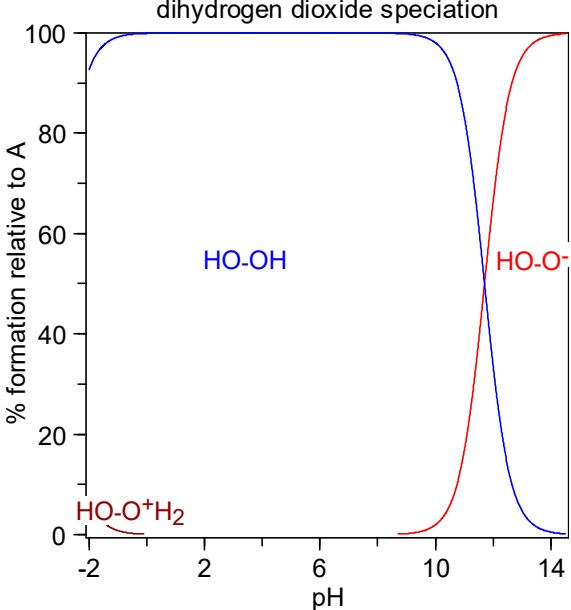

**Figure 6.** Diagram of dihydrogen dioxide distribution: percent species formation calculated with the HySS program (Alderighi et al., 1999) [47] for a 10 millimolar solution of $H_2O_2$ ($pK_{a1} = -3.1$, $pK_{a2} = 11.7$).

*3.2. Reaction of PMG and H$_2$O$_2$*

It is generally known that the oxidation potential of H$_2$O$_2$ greatly increases during UV irradiation (Mierzwa et al., 2018, and references cited therein) [50] as well as in the presence of metal ions (Figure 7) [50–55].

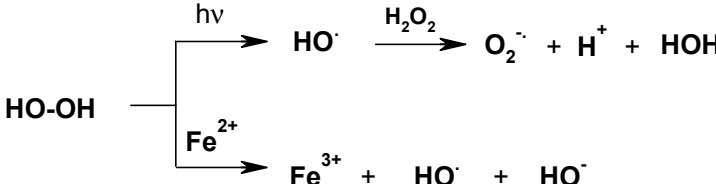

**Figure 7.** Activation of dihydrogen peroxide via the generation of radicals.

Therefore, we assumed that the reaction between PMG and dihydrogen peroxide consumed either the molecular form of H$_2$O$_2$ in the absence of irradiation, or hydroxide and peroxide radicals during UV irradiation or in the presence of Fenton reagents. The results of PMG degradation in both modes are illustrated in Figure 8.

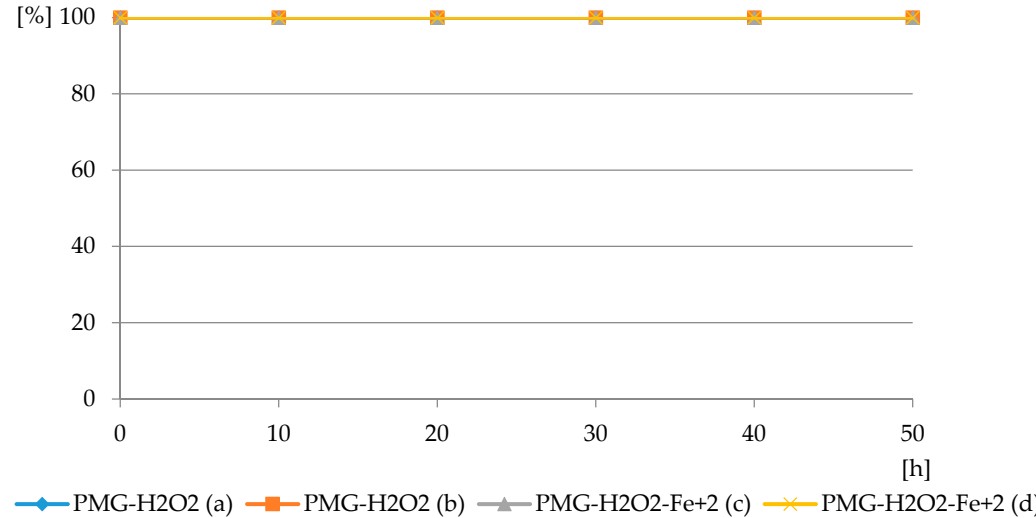

**Figure 8.** The profiles of PMG consumption in reaction with H$_2$O$_2$ in PMG-H$_2$O$_2$ and PMG- H$_2$O$_2$-Fe$^{2+}$ systems at pH = 3.5: PMG:H$_2$O$_2$ = 1:5 (PMG-H$_2$O$_2$ (a)) and 1:10 (PMG-H$_2$O$_2$ (b)); PMG:H$_2$O$_2$:Fe$^{2+}$ = 1:10:0.05 (PMG-H$_2$O$_2$-Fe$^{2+}$ (c)); and 1:10:0.1 (PMG-H$_2$O$_2$-Fe$^{2+}$ (d)) (residual PMG (%) vs. exposure time (h)).

The reaction of PMG with H$_2$O$_2$, both with H$_2$O$_2$ and H$_2$O$_2$-Fe$^{3+}$, did not occur at the applied pH = 3.5, at which PMG exists mainly in the H$_2$L$^-$ protonated on nitrogen form (Figures 3 and 4) and hydrogen peroxide mainly in H$_2$L forms (Figures 5 and 6). Therefore, we assumed that the protonation of the amino function in PMG efficiently reduces the interaction of PMG and H$_2$O$_2$ (no trace of P-C bond splitting was observed in a 48-h period) (Figure 8). However, during the irradiation of aqueous solutions of PMG (in the form of the herbicide Roundup) and H$_2$O$_2$ (1:5), for a pH range of 2 $\leq$ pH $\leq$ 12, the splitting of the P-C bond of PMG was observed, to an extent dependent on the pH of the applied solution (Figures 9 and 10). Is worth noting that the irradiation of aqueous PMG solution without H$_2$O$_2$ during a 48-h period did not exhibit any sign of PMG decomposition (100% of PMG).

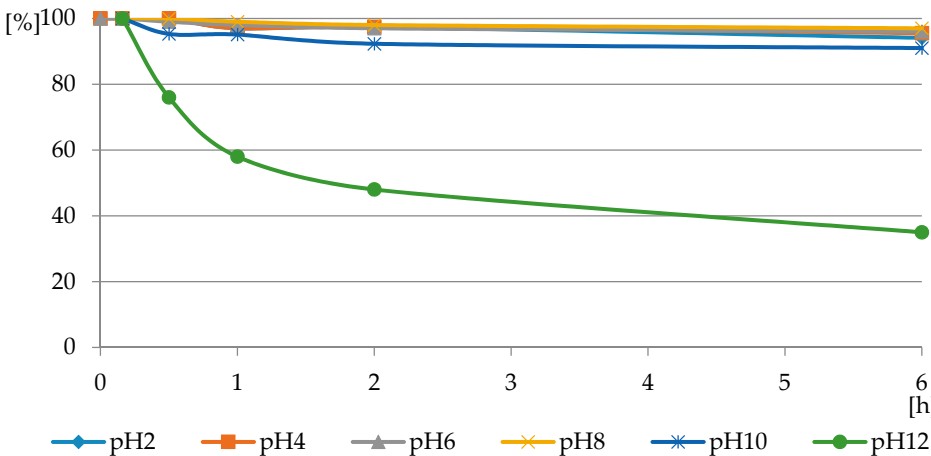

**Figure 9.** The profile of PMG consumption in reactions with $H_2O_2$ in PMG-$H_2O_2$-(UV) systems (with UV irradiation) carried out at a pH range of $2 \leq pH \leq 12$ and a temperature of 25 °C (residual PMG (%) vs. exposure time (h)).

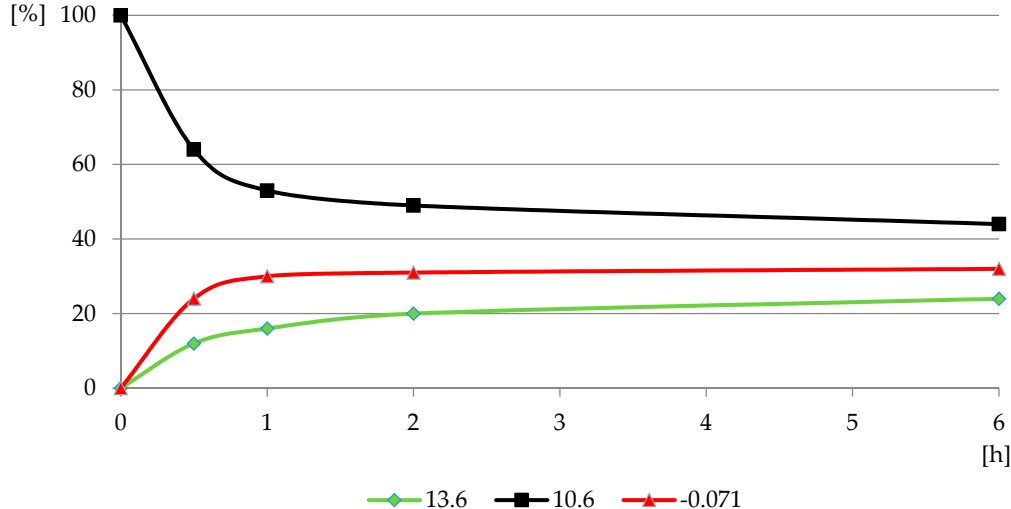

**Figure 10.** The profile of PMG reaction with $H_2O_2$ in PMG-$H_2O_2$-(UV) systems (UV; 25 °C) carried out at pH = 12 (relative P-compounds contribution ($Gly^P$: 13.6 ppm, PMG: 10.6 ppm; $P_i$: −0.071 ppm) (%) vs. exposition time (h)).

The residual PMG quantities were calculated from the corresponding $^{31}P$ NMR spectra using Equation (1):

$$PMG = \frac{S_{(PMG)}}{S_{(PMG)} + S_{(R-P)} + S_{(Pi)}} \times 100\%, \tag{1}$$

where $S_{(PMG)}$, $S_{(R-P)}$, and $S_{(Pi)}$ are the $^{31}P$ signals corresponding to PMG, phosphonic acids, and inorganic phosphate, respectively.

The $^{31}P$ NMR spectra of the degradation mixtures of PMG-$H_2O_2$-(UV) (UV; 25 °C) recorded for reactions carried out at pH = 2, 8, 10, and 12 are presented in Figure 11. For the identification of the reaction products of PMG-$H_2O_2$-(UV) mixtures, we recorded the $^{31}P$ NMR spectra of PMG potential degradation products. The chemical shifts ($\delta(^{31}P)$) of these compounds are listed in Table 6.

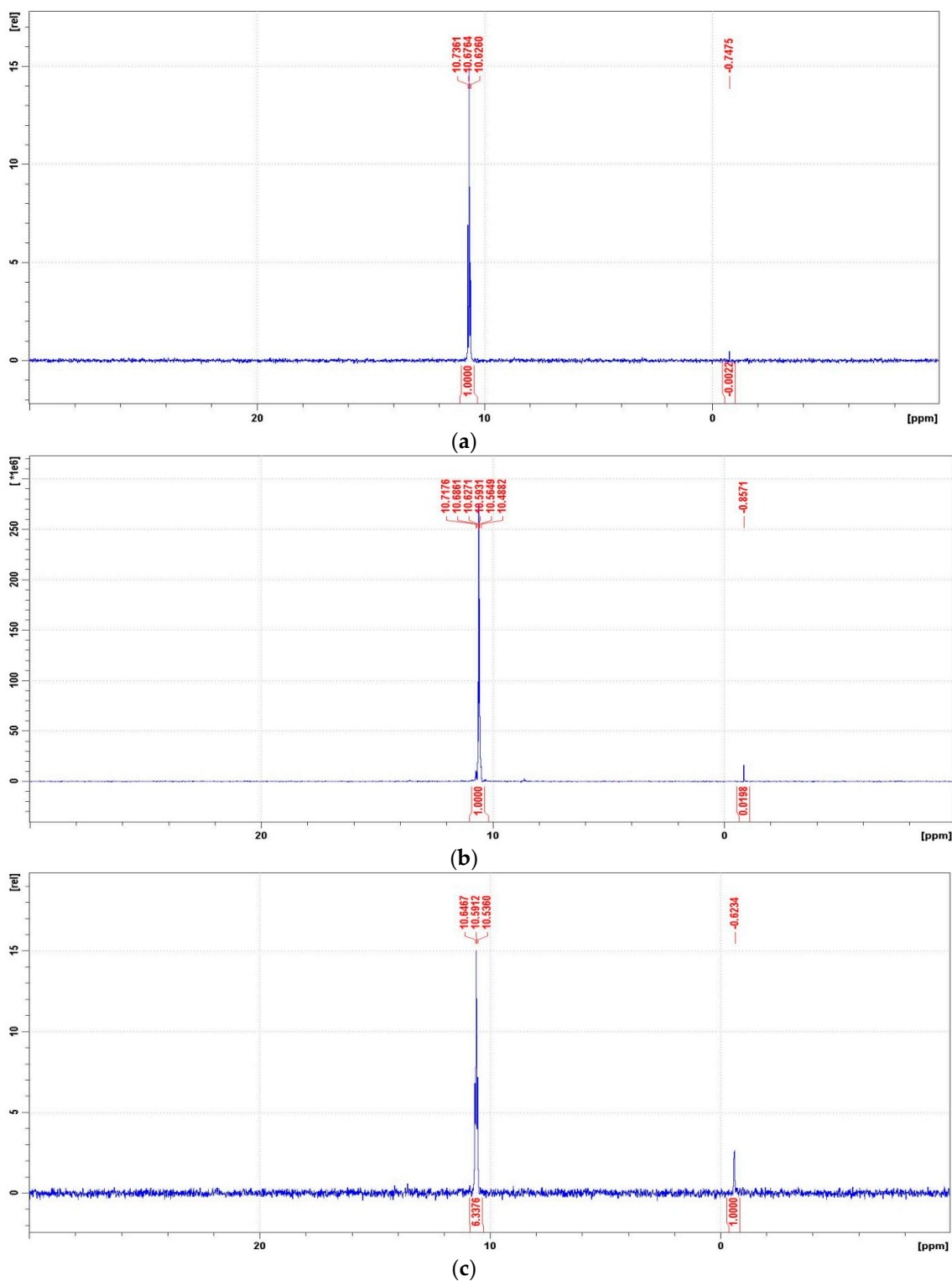

**Figure 11.** *Cont.*

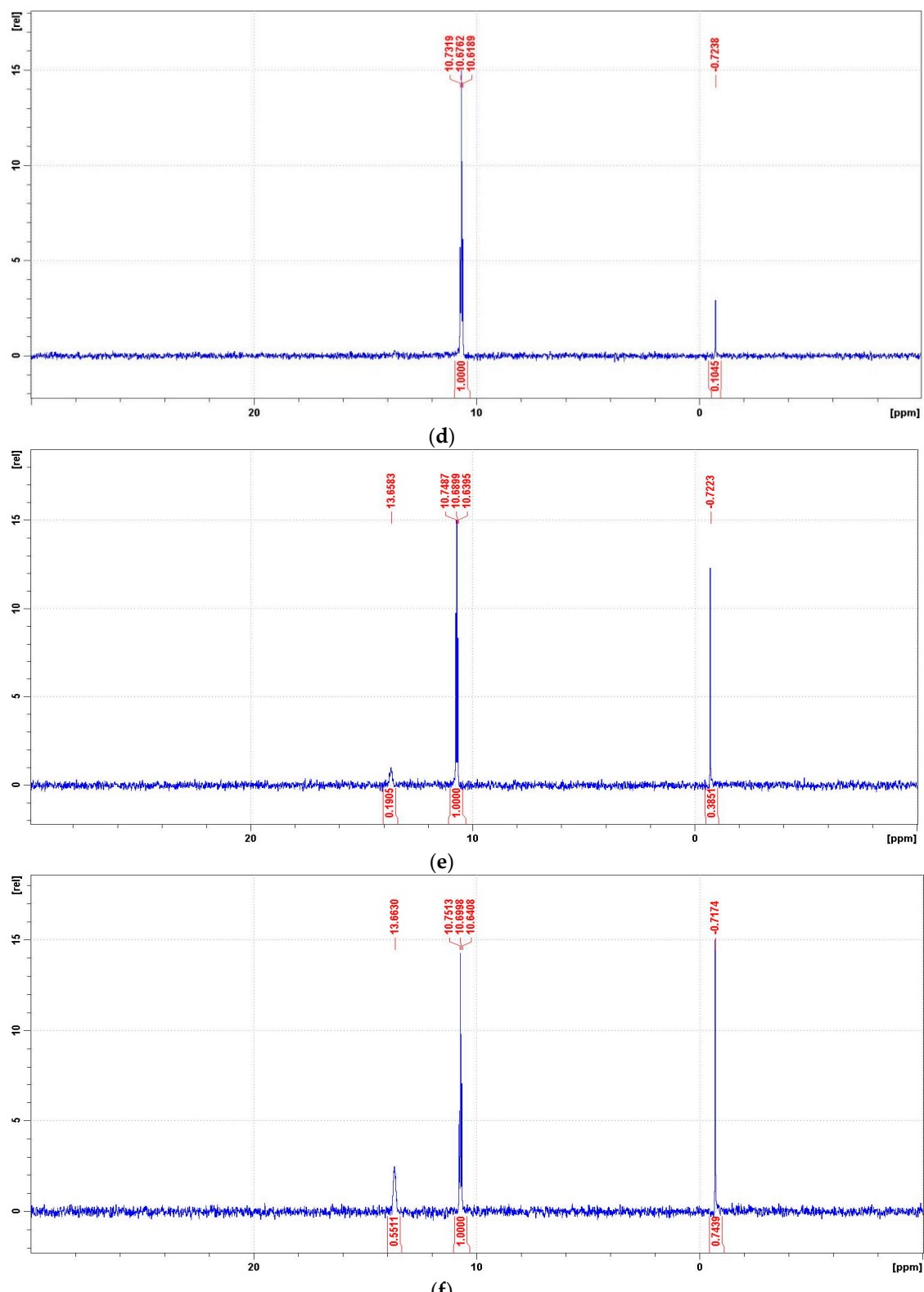

**Figure 11.** Representative $^{31}$P NMR spectra of the degradation mixtures of PMG-H$_2$O$_2$-(UV) carried out at pH = 2–12 (UV; 25 °C), recorded after a given reaction time. (**a**) $^{31}$P NMR spectrum of the reaction mixture recorded before radiation (dissolved in 2 M H$_2$SO$_4$/D$_2$SO$_4$/); (**b**) reaction mixture after UV radiation at pH = 2 (reaction time 360 min); (**c**) reaction mixture after UV radiation at pH = 8 (reaction time 360 min); (**d**) reaction mixture after UV radiation at pH = 10 (reaction time 360 min); (**e**) reaction mixture after UV radiation at pH = 12 (reaction time 30 min); (**f**) reaction mixture after UV radiation at pH = 12 (reaction time 360 min).

**Table 6.** $^{31}$P Chemical shifts ($\delta$(ppm)) of PMG and its potential degradation products in acidic and basic solutions.

| **2 M HCl** | | | | | | | |
|---|---|---|---|---|---|---|---|
| Comp. | PMG | Gly$^P$ | Me-Gly$^P$ | Me$_2$-Gly$^P$ | Me-PO$_3$H$_2$ | H$_3$PO$_4$ | H$_3$PO$_3$ |
| $\delta$ ($^{31}$P) (ppm) | 10.6 | 13.9 | 11.4 | 9.4 | 30.7 | $-0.47$ | 5.15 |
| **2 M KOH** | | | | | | | |
| Comp. | PMG | Gly$^P$ | Me-Gly$^P$ | Me$_2$-Gly$^P$ | Me-PO$_3$H$_2$ | H$_3$PO$_4$ | H$_3$PO$_3$ |
| $\delta$ ($^{31}$P) (ppm) | 16.3 | 19.3 | 16.0 | 15.0 | 20.5 | 5.4 | 3.2 |

$^{31}$P $\delta$(ppm): in 2 M HCl solutions (protonated forms of P-acids); in 2 M KOH solutions (deionized forms of P-acids).

The results of $^{31}$P NMR investigations on PMG degradation with H$_2$O$_2$ are shown graphically in Figure 12.

**Figure 12.** Possible degradation paths of PMG (reaction time: 48 h).

## 4. Conclusions

The data presented suggest the PMG inertness toward H$_2$O$_2$ in the modes without UV irradiation, both with (PMG-H$_2$O$_2$-Fe$^{2+}$) as well as without Fe$^{2+}$ catalyst (PMG-H$_2$O$_2$). The data considering the reaction modes of PMG with H$_2$O$_2$ under UV irradiation (PMG- H$_2$O$_2$-(UV)) exhibit the slow degradation of PMG at $2 \leq$ pH $\leq 10$, which becomes faster at pH = 12. The analysis of the $^{31}$P NMR spectra of PMG-H$_2$O$_2$ reaction mixtures obtained for reactions carried out at $2 \leq$ pH $\leq 10$ indicate the presence of initial PMG and H$_3$PO$_4$, and the mixture of PMG, Gly$^P$, and H$_3$PO$_4$/H$_x$PO$_4{}^{3-x}$ for reactions carried out at pH = 12. The results suggest the slow formation of an intermediate PMG $\times$ H$_2$O$_2$ phase in the first stage of degradation which decomposes very fast (no intermediates were observed in the $^{31}$P NMR spectra) by the scission of the P-C bond of PMG and the subsequent release of phosphoric acid/phosphate ion.

Recapitulating, in the experiments carried out without UV radiation we observed:

- full stability of PMG in reaction with $H_2O_2$ (48 h);
- full stability of PMG in reaction with $H_2O_2/Fe^{2+}$ (48 h).

In the experiments carried out with UV radiation (PMG-$H_2O_2$-(UV)), the P-C rapture type of PMG degradation was observed, the extent of which was dependent on the applied pH of the reaction mixtures. As a result, for the reactions run at $2 \leq pH \leq 10$, the partial formation of phosphoric acid/phosphate ions (PMG $\rightarrow$ PMG + $H_3PO_4$/$H_xPO_4^{3-x}$) was observed, whereas for reactions run at pH = 12, mixtures of PMG, $Gly^P$, and $PO_4^{3-}$ were found.

We did not observe:

- any formation of nitrone-type derivatives (see [56–60]);
- the formation of Me-$Gly^P$ ($Sar^P$) or Me-P(O)(OH)$_2$.

**Author Contributions:** M.H.K. designed the research study and contributed to the data interpretation and to the manuscript drafting and revisions and was involved in the concept of the research study, analyzed the data, and contributed to writing the manuscript. R.Ż. participated in the publication preparation. Z.M. performed experiments. P.U. recorded N.M.R. spectra and analyzed the experimental data.

**Funding:** This research was funded by the Polish Ministry of Science and Higher Education within statutory research work carried out in 2018 at the Textile Research Institute, Łódź, Poland.

**Conflicts of Interest:** The authors declare no conflict of interest.

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
