# Peer review of "31P NMR Investigations on Roundup Degradation by AOP Procedures"

_water, doi:10.3390/w11020331_

Round 1

Reviewer 1 Report

Present manuscript describes studies on oxidative degradation of PMG with hydrogen peroxide under various conditions. Overall it is a nice piece of work which is related to water treatment and water quality. However, it sounds very descriptive and the results obtained by authors are not put into perspective with what is known. In other words, in the “results and discussion” section, discussion is missing. Also, oftentimes, the reasons behind the choice of reagents and reaction conditions is not clear. So, for example, why the oxidation reaction of PMG with hydrogen peroxide was tested only at pH =3.5 and not at other pH? Below are some other minor comments:

All graphs: Please put axis description and units near axis. For example: Time, h; Residual PMG, % etc.

Line 2: “utilization” does not seem to be the most appropriate word. Please replace it by more suitable alternative (like degradation).

Line 10: delete word “applied”

Line 21: Introduce abbreviation AAA on the first appearance.

Line 36-37: change “investigate these process, using 31P NMR monitoring of the PMG degradation processes” to “investigate these degradation processes by using 31P NMR monitoring”

Line 51: Table 2 footnote: correct “JUPAC to “IUPAC”

Line 54: change “Diisopropylphosphite were”  to “Diisopropylphosphite was”

Line 69 and below: “was homogeneous at 1H NMR and 31P NMR” does not sound well. Please consider replacing it by: “1H NMR and 31P NMR spectra of obtained Glyp (8.1 g, 0.072 mole, 72%) were identical to those previously reported. (add ref)”

Figure 1: replace” Reactor used for Roundup utilization by” by “Reactor used for Roundup degradation by “

Line 114-115: “samples of PMG (12 mmol), contained in Roundup Ultra 170 SL Herbicide (16 ml)”. Does this mean that 16 mL of Roundup solution contained 12 mmol of PMG? Please rephrase it to make it more clear.

Table 3: Please make the table more clear. What is shown in the column between columns 10 M H2O2 and FeSO4? What is the volume indicated in PMG column entry 1?

Figure 4 and the text associated therewith: The reason of choosing pH value 3.5 for this experiment is not clear. Assumption that “protonation of the amino function in PMG efficiently decrease an interaction of PMG and H2O2” is not supported by data. Does PMG reacts with H2O2 or H2O2/Fe(II) at higher pH?

Figure 6: It seems that y axis here represent relative signal intensity in 31P NMR spectra, but not %PMG. Please correct.

Figures 7, 8: NMR spectra are too little. It is hard to read. Please zoom them and use larger font.

Lines 206-207: “The data presented suggest that PMG are inert toward H2O2” this has been shown only for one pH value. Please correct appropriately.

Lines 207-208: “The analysis of 31P NMR spectra of AAP-H2O2 reaction mixtures”. Did the authors mean PMG-H2O2 reaction mixtures? Please correct accordingly.

Author Response

Reviewer 1; Review   Report (Round 1)

Query

No.

Line/ Tab./Fig.

Before correction

After correction

1

L. 2

31P NMR investigations on Roundup utilization by AOP procedures

31P NMR investigations on Roundup degradation by AOP procedures

2

L. 10

the splitting of C-P was observed in the ratios   dependent on the applied   pH of the reaction mixture.

the splitting of C-P was observed in the ratios   dependent on the pH of the reaction mixture.

3

L. 21

EC   2.5.1.19) – involved in the bio-synthesis of AAA substrates …

EC 2.5.1.19) – involved in the bio-synthesis of aromatic amino acids (AAA) substrates …

4

L. 36/37

… to investigate these process, using 31P   NMR monitoring …

… to investigate these process by using 31P   NMR monitoring …

5

L. 51

a/Applied names were in accordance with the JUPAC rules

a/Applied names were in accordance with the IUPAC rules

6

L. 54

Diisopropylphosphite were purchased from ACROS OrganicTM.

Diisopropylphosphite was purchased from ACROS OrganicTM.

7

L. 69

was homogeneous at 1H NMR and 31P NMR

we supported these data with elemental analysis; 31P   NMR data are listed in Table 6

8

Fig. 1

Reactor used for Roundup utilization   by

Reactor used for Roundup degradation   by

9

L. 114/

115

“samples of PMG (12 mmol), contained in Roundup   Ultra 170 SL Herbicide (16 ml)”. Does this mean that 16 mL of Roundup   solution contained 12 mmol of PMG? Please rephrase it to make it more clear.

…, samples of PMG contained in Roundup Ultra 170 SL Herbicide (16 ml; 12   mmol of PMG),..

10

L. 206/

207

“The data presented suggest that PMG are inert   toward H2O2” this has been shown only for one pH value. Please correct   appropriately.

The reactions of PMG with H2O2   were run under UV irradiation, at ambient temperature for 48h….. The data presented suggest   that PMG are inert toward H2O2 without UV irradiation,   (pH=3.5)

11

L. 207/

208

“The analysis of 31P NMR spectra of AAP-H2O2   reaction mixtures”. Did the authors mean PMG-H2O2 reaction mixtures? Please   correct accordingly.

The analysis of 31P NMR spectra of PMG-H2O2   reaction mixtures …

12

All graphs

Please put axis description and units near axis. For   example: Time, h; Residual PMG, % etc.

OK

13

Table 3:

Please make the table more clear. What is shown in   the column between columns 10 M H2O2 and FeSO4?   What is the volume indicated in PMG column entry 1?

The table 3 have been corrected

(formatting error).

14

Figure 4 and the text   asso-ciated therewith:

The reason of choosing pH value 3.5 for this   experiment is not clear. Assumption that “protonation of the amino function   in PMG efficiently decrease an interaction of PMG and H2O2” is not supported   by data. Does PMG reacts with H2O2 or H2O2/Fe(II) at higher pH?

The solutions PMG-H2O2-H2O-Fe(II) with higher   concentration of Fe(II) exhibited pH 3.5. For unifications, other applied   solutions (pH ~ 4) were acidified to this value of pH, since 31P  shifts of P-acids strongly depend on the pH   [see Refs. 27 and 28].

15

.., why the oxidation reaction of PMG with hydrogen   peroxide was tested only at pH =3.5 and not at other Ph.

At higher pH precipitation of some type of colloids   [presumably PMG×Fe(II) + Fe(OH)2] appeared, which, in spite of prior   centrifugations, caused widening of 31P signals

16

Figure 6:

It seems that y axis here represent relative signal   intensity in 31P NMR spectra, but not %PMG. Please correct.

Figure   6. The profile of   PMG reaction with H2O2 in systems PMG-H2O-H2O2   (UV, temp. 25°C) at pH = 12; temp. 25°C {relative P-compound contribution (GlyP: 13.6 ppm, PMG:   10.6 ppm; Pi: -0.071)   [%] vs exposition time [h]}

17

Figures

7, 8:

NMR spectra are too little. It is hard to read.   Please zoom them and use larger font.

The figures 7 and 8 have been enlarged.

Reviewer 2 Report

The authors tried to find Roundup utilization by various AOP procedures. The ms is very interesting and from my point of view deserves publication. Nevertheless, several references should be cited in the Introduction section to elaborate about the two forms (glyphosate and Roundup), for example: Toxics 2018, 6, 2; doi:10.3390/toxics6010002 and Int. J. Environ. Res. Public Health 2018, 15, 1117; doi:10.3390/ijerph15061117. The authors may also elaborate in the Discussion section and not just to mention what they found in bullets. The authors may state about the aim of their study more clearer and to mention in the Discussion section what exactly they found and what is the interpretation of these findings. Last some figures are very hard to follow and need more explanations.

Author Response

Reviewer 2 ; Review   Report (Round 1)

Q.

Author’s remark

Text in original version

Text after correction

1

Nevertheless, several   references should be cited in the Introduction section to elaborate about the   two forms (glyphosate and Roundup), for example: Toxics 2018, 6, 2;   doi:10.3390/toxics6010002 and Int. J. Environ. Res. Public Health 2018, 15,   1117; doi:10.3390/ijerph15061117.

Also   the World Health Organization’s International Agency for Research on Cancer   concluded that glyphosate is “probably   carcinogenic to humans.” [13] and recently Alleeva et al. [14]   demonstrated that PMG, even at low doses, induce DNA damage responses.

Also the World Health   Organization’s International Agency for Research on Cancer concluded that   glyphosate is “probably carcinogenic to   humans.” [13] and Anifandis et al. [14,15] demonstrated that   glyphosate/PMG, induce DNA fragmentation.

2

The authors may also   elaborate in the Discussion section and not just to mention what they found   in bullets.

The Conclusion part   has been extended. In this new form it answers the Referee query.

3

The authors may state   about the aim of their study more clearer and to mention in the Discussion   section what exactly they found and what is the interpretation of these   findings

4

Last some figures are   very hard to follow and need more explanations.

All figures have been   corrected.

Reviewer 3 Report

The manuscript demonstrates the application of phosphorus 31 NMR spectroscopy in characterizing oxidative degradation of glyphosate by hydrogen peroxide combined with UV irradiation or Fe ions. While the research investigation has its importance, presentation of the results needs to be improved especially on the quantitative aspect. Below are my comments that the authors may want to address in the revised manuscript.

·        Based on my understanding of the manuscript, it seems that the term “utilization” can be replaced by “degradation” or equivalent, to avoid confusion.

·        The English grammar needs to be checked carefully, especially on the use of punctuation and single/plural forms. Use of an English editing service is highly recommended.

·        Full terms should be given when abbreviations are used the first time, for instance, line 20, EPSP, Line 21, AAA substrates. In the title of table 1,  Glyp is a term that was not defined prior to its appearance at this point. The name used for the same compound should be also consistent, for example, hydrogen peroxide vs hydrogen dioxide.

·        The caption for figure 1 should be revised and contains complete sentences, except the title.

·        It would be in a  more logical order if representative 31P NMR spectra were presented first to show how oxidation is observed and studied (Figure 8) before comparing reactions under different conditions (Figures 4, 5, and 6)

·        How did the authors determine the percent degradation of PMG by 31P NMR quantitatively? It is not described anywhere in the manuscript.

·        In Figure 6, what do the three values/curves indicate? The information can’t be found in the caption or the main text.

·        The resolution of NMR images in Figure 7 and Figure 8 needs to be improved. 

Author Response

Reviewer 3; Review   Report (Round 1)

Sugg.#

Line

Suggestions

After   correction

1

L.   2. &

Fig.   1

…the term “utilization” can be replaced by   “degradation” or equivalent, ….

The term the term “utilization” has been replaced by   degradation (see   line 2 and Fig. 1)

2

20/21

Full terms should be given when abbreviations are   used the first time, for instance, line 20, EPSP, line 21 - AAA

…. – 5-enolpyruvylshikimate-3-phosphate (EPSP) synthetase (EC   2.5.1.19) – involved in the bio-synthesis of aromatic amino acids (AAA) substrates

3

Table   1

In the title of table 1,  Glyp is a term that   was not defined prior to its appearance at this point.

Explanation is given as the footnote of the Table: GlyP –  see Scheme 1.

4

Lines:   111, 118, 148;

Fig.   3; Sch. 3, 4

The name used for the same compound should be also   consistent, for example, hydrogen   peroxide vs hydrogen dioxide.

We unified the names of H2O2   as: dihydrogen peroxide

5

It would be in a  more logical order if   representative 31P NMR spectra were presented first to show how   oxidation is observed and studied (Figure 8) before comparing reactions under   different conditions (Figures 4, 5, and 6)

Figures 7 and 8 are 31P NMR illustrations   - supplemental to Figures 5 and Fig. 6.

6

How did the authors determine the percent   degradation of PMG by 31P NMR quantitatively? It is not described   anywhere in the manuscript.

These are taken from the spectra:

Where S(PMG), S(R-P) and S(Pi)   present

the areas of the 31P signal corresponding   to PMG; phosphonic acids, and inorganic phosphates, respectively.

7

In Figure 6, what do the three values/curves   indicate? The information can’t be found in the caption or the main text.

Figure 6. The profile of PMG   reaction with H2O2 in systems PMG-H2O-H2O(UV, temp. 25°C) at pH = 12; temp.   25°C {relative   P-compounds contribution (GlyP: 13.6 ppm, PMG:   10.6 ppm; Pi: -0.071) [%] vs exposition time [h]}

8

The resolution of NMR images in Figure 7 and Figure   8 needs to be improved. 

The spectra have been enlarged

9

The caption for figure 1 should be revised and   contains complete sentences, except the title.

The title of Fig. 1 has been corrected

Round 2

Reviewer 1 Report

Please correct diapasons like "2>pH>12" to "2<pH<12" throughout the paper. 

Author Response

Table 1. The   list of Referrer’s Suggestions/Queries & Answers (Referee #1)

Q

L

Suggestion

Before   correction

After correction

1

16

Please correct diapasons like   "2>pH>12" to "2<pH<12" throughout the paper

2 ≥ pH ≥ 12

2 ≤ pH ≤ 12

193

2 ≥ pH ≥12

2 ≤ pH ≤ 12

198

pH2 ≥ pH ≥12

2 ≤ pH ≤ 12

226

2≥pH≥10

2 ≤ pH ≤ 10

227

2≥pH≥10

2 ≤ pH ≤ 10

237

2≥pH≥8

2 ≤ pH ≤ 10

Reviewer 2 Report

The authors made a very good revision and have answered the referee's suggestions

Author Response

Dear Sir,

Thank you for taking your time for the review.

With regards,

Marcin H. KUDZIN

Reviewer 3 Report

Line 12, …5÷10 excess of hydrogen dioxide…, I believe the authors meant  5~10 or 5-10.

Line 16~17, the splitting of C-P 16 was observed in the ratios dependent on the applied pH.., ratios of what? It should be described clearly.

Scheme 2 has two figures between line 142 and 145 that are identical but of different sizes/arrangements. Only one image is needed.

 Figure 7 and Figure 8 show the 31P-NMR  spectra of the degradation mixtures of PMG-H2O-H2O2-UV after 6 hours at pH 10 and pH 12. The two sets of images are exactly the same. The degradation percentages of PMG, therefore, would be the same at pH 10 and pH 12, which is very different from what’s shown in Figure 5 (where more than 60% degradation at pH 12 compared to less than 10% at pH 10). I believe the images in Figure 7 are not representative of the reaction at pH 10.

The grammar in the text between line 232 to line 235 should be checked and corrected.  Also in line 233 and 234, the word ”stability “should have an adjective, it that “high stability” or “low stability”?

Author Response

Table 2. The list of Referrer’s   Suggestions/Queries & Answers  (Referee   #3)

Q

L

Suggestion

Before correction

After correction

1

12

5÷10 excess of hydrogen dioxide…,

 I believe the authors meant    5~10 or 5-10

5÷10 excess of hydrogen dioxide

5-10 excess of hydrogen   dioxide

2

16-17

the splitting of C-P 16 was   observed in the ratios dependent on the applied pH.., ratios of what?

 It   should be described clearly.

In this mode of PMG oxidation, the splitting of C-P bond was   observed in the ratios dependent on the applied pH of the reaction mixture.

3

Sch. 2

has two figures between line 142 and 145 that   are identical but of different sizes/arrangements.

Only one image is needed.

We enlarged the size of   Scheme 2 to size of other figures.

We have chosen the scheme of   enlarged size.

4

Figure 7

and

Figure 8 

Figure 7 and Figure 8 show the 31P-NMR    spectra of the degradation mixtures of PMG-H2O-H2O2-UV after 6 hours at pH 10   and pH 12. The two sets of images areexactly the same. The degradation   percentages of PMG, therefore, would be the same at pH 10 and pH 12, which is   very different from what’s shown in Figure 5 (where more than 60%   degradationat pH 12 compared to less than 10% at pH 10).

I believe the images in Figure 7 are not   representative of the reaction at pH 10.

Figures 7 & 8 (obviously the same) have been   substituted by the set of representative figures, assigned now as Figure(s)   7.

5

232

The grammar in the text between line 232 to   line 235 should be checked and corrected. 

OK

6

233,234

Also in line 233 and 234, the word ”stability   “should have an adjective, it that “high stability” or “low stability”?

·           Stability   of PMG in reaction with H2O2 (48 h);

·           Full   stability of PMG in reaction with H2O2 (48 h);

234

·           Stability   of PMG in reaction with H2O2/Fe2+ (48 h).

·           Full   stability of PMG in reaction with H2O2/Fe2+   (48 h).
